# Gender and Sex in Medical Practice: An Exploratory Study on Knowledge, Behavior, and Attitude among Sicilian Physicians

**DOI:** 10.3390/ijerph20010827

**Published:** 2023-01-01

**Authors:** Giuseppina Campisi, Fortunato Buttacavoli, Massimo Attanasio, Mirella Milioto, Silvia Radosti, Salvatore Amato, Vera Panzarella

**Affiliations:** 1University Hospital of Palermo, University of Palermo, 90127 Palermo, Italy; 2Medical Council of Palermo, Gender Medicine Board, 90146 Palermo, Italy; 3Department of Surgical, Oncological and Oral Sciences (Di.Chir.On.S.), University of Palermo, 90127 Palermo, Italy; 4Department of Economic, Business and Statistic Sciences (SEAS), University of Palermo, 90127 Palermo, Italy

**Keywords:** gender-specific medicine, sex-specific medicine, precision medicine, post-graduate medical education, gender, sex, cardiovascular disease, tobacco cessation

## Abstract

Personalized medicine is a new paradigm in health care, and the concept of socio-cultural gender, as opposed to biological sex, emerged in several medical approaches. This exploratory study aimed to investigate the knowledge of sex and gender in clinical medicine among Sicilian physicians. Data collection was based on an online survey sent to the members of the Medical Councils of Sicily (Italy). The questionnaire included nine specific items about awareness and attitudes regarding gender medicine and its importance in clinical practice. 8023 Sicilian physicians received the solicitation e-mail and only 496 responded. Regarding the knowledge of gender medicine, 71.1% of participants stated that they know it, while 88.5% believe that gender medicine should be included in training programs. Similarly, a high percentage (77.6%) would like to keep up to date on this topic. Physicians sampled seem to understand the importance of gender medicine principles, although their experience of some gender issues (i.e., sex disparities in acute cardiovascular care and smoking cessation strategies) is low (55.44% and 21.57%, respectively). The results of this exploratory study should encourage facing the gender medicine gap in the current curricula of health professionals and should implement the transitional value of sex and gender principles in the clinical setting.

## 1. Introduction

Gender and sex medicine is defined as the practice of medicine in which sectorial determinants are considered, such as mainly biology (by genes and hormones) and social roles (gender) with implications for the prevention, screening, diagnosis, and treatment of human beings.

Commonly labeled as “gender medicine”, it is a fundamental approach to improve the quality of care and to reduce the gender gap among patients. Unfortunately, the androcentric bias, prevalent both in healthcare and medical research, still has major impacts on health inequalities between men and women [1]. These disparities are known in epidemiology, pathophysiology, clinical manifestations, disease progression and response to treatment of numerous chronic diseases: heart disease, cancers, chronic pulmonary disease, stroke, Alzheimer’s disease, diabetes, chronic kidney, liver diseases and depression [2]. Despite several international policies finalized to include sex and gender in medical research, the influence of sex and gender on healthcare continues to be underestimated, understudied, and underutilized in clinical practice [3]. The still scarce attitude and knowledge of clinicians and researchers regarding the importance of sex and gender in medicine represent the very key barrier to proposing effective efforts to promote gender equity at all levels of the biomedical enterprise [4]. With the approval of law 3/2018 (namely “Delegation to the Government concerning clinical trials of medicinal products and provisions for the reorganization of the health professions and the healthcare officials of the Ministry of Health”) [5], according to the regulation of the European Parliament [6], for the first time in Europe, the insertion of “gender” is guaranteed in all medical specialties and clinical trials along with the definition of diagnostic-therapeutic paths, research, training and dissemination to all health professionals and citizens. Particularly, this law required a plan aimed at spreading gender medicine to ensure the quality and appropriateness of services provided throughout the nation.

According to this healthcare disposition, a technical-scientific panel of experts in gender medicine will be involved in the creation of clinical networks to achieve general and specific objectives for 4 definite areas: (a) clinical pathways, (b) research and innovation, (c) professional training courses, (d) communication and information; specifically, the aim is “to promote awareness of gender differences in healthcare to transfer the knowledge and skills gained in professional activities” [7].

To date, to the best of our knowledge, no information is available on Sicilian physicians and their knowledge about gender medicine in research activity and clinical practice. By collecting the opinions of Sicilian physicians on these topics, this study examined their knowledge, behavior, and attitude on the role of gender medicine and the need for dedicated regional educational programs aimed at promoting the awareness of sex and gender differences in health care.

## 2. Materials and Methods

### 2.1. Study Design, Survey Target Sample and Dissemination Strategy

This study was exploratory with the specific aim to investigate the knowledge of sex and gender-related differences in a sample of generic and specialized Sicilian physicians. For this reason, a short anonymous online survey was administered to all physicians affiliated with the Medical Council Registers of Sicily (Italy), (https://www.ordinemedicipa.it/, accessed on 4 November 2021) [8]. Between 20 October 2021 and 31 July 2022, the survey was circulated strategically among all members of the Medical Councils of Sicily (Italy), by sending a specific e-mail, containing details and objectives of the study. All the physicians had to give mandatory information including age (years), gender and medical specialty/subspecialty. The clusterization of medical specialties was made by the affinity of related disciplines; in particular, each medical specialty was grouped for convenience criteria and according to different axes: diagnostic/therapeutic, organ-based, or field-based (as indicated in Table 1). 8023 Sicilian physicians received the e-mail soliciting to answer the online questionnaire.

### 2.2. Questionnaire

The online questionnaire was commissioned by the Medical Council of Palermo (Sicily, Italy) to the University of Palermo, Italy (MA, GC, VP, FB). Other than the mandatory demographic questions (age, gender and specialty/sub-specialty) the short survey included 9 closed-ended questions concerning the areas of knowledge and attitudes about gender medicine and the importance of sex in disease prevalence, manifestation, and response to treatment (Table 2). Participants were recruited by the distribution of an approved e-mail, specifically concerning the study and with the link to the online questionnaire (https://sondaggi.mediawam.it/forms/indagine-sulla-medicina-di-genere-rivolta-ai-medici-iscritti-all-ordine-di-palermo, accessed on 1 October 2021) [9]. The link has been active for 9 months (October 2021/July 2022 and the survey required approximately ten minutes to be completed by each responder.

### 2.3. Ethical Considerations

The study was conducted under the approval of the Medical Council of Palermo (minutes approval date: 24 September 2021). No compensation or incentive was offered to the surveyed physicians and their participation was strictly anonymous and voluntary; the information collected has been protected as required by the current law on the processing and protection of personal data.

### 2.4. Statistical Analysis

Descriptive statistics were carried out. Specifically, all categorical variables were expressed as counts and percentages, differentiating by gender (women vs. men) and medical specialties (Group I to VII). Given the exploratory nature of the study, no preliminary sample measurements, or comparative evaluation of the statistical difference between the categorical variables were carried out.

## 3. Results

Of the 8023 physicians registered with the Medical Council Registers of Sicily, 496 responded; most came from Palermo (486; 98%) and belonged to Group VI of medical specialties (258; 52%). The highest frequency of respondents came from women (261; 52.6%), among these, most were aged over-50s (Table 3).

All answers details of the 9 items, differentiated by gender (woman and men), are shown in Table 4.

Concerning the knowledge of gender medicine (ITEM#1: “Do you know what gender medicine deals with?”), 71.2% (353/496) of respondents (180 women and 173 men) reported that they knew the fields of interest of gender medicine. One hundred forty-three (81 women and 62 men, for a total of 28.8% of respondents) answered “No” (Table 2). Regarding medical specialties, most positive responses occur in Group II for men (100%) and Group VI for women (97.2%) (Appendix A).

Concerning the usefulness of gender medicine (ITEM#2: ”In clinical practice, does the knowledge of the differences between sex and gender improve the ability to treat patients?”), 87.9% (436) of respondents (224 women and 212 men) recognized that knowing the differences between sex and gender improve the health care ability. Forty-five (9%) answered “Maybe”, and fifteen (3%) “No”. No distinctions were detected among medical specialties (Appendix A).

For sex-gender selectivity of clinical trials in medicine (ITEM#3: “In your opinion, does most of the medical knowledge comes from studies conducted mainly on men?”), only 42% of respondents (110 women and 98 men) recognized that clinical trials were designed for men. 58% of the respondents replied “No” or “I do not know” with no differences between men and women.

Regarding medical specialties, strong differences emerged between men and women in specific groups, with the highest percentage of women who responded positively to this question in Group III (76.4%) and the totality of men (100%) in Group II who answered that they did not know. Moreover, 50% of men in Group I answered ‘Yes’, compared with 14.2% of women in the same group. There were no differences between gender for the other Group (Appendix A).

As regards the importance of differentiating pharmacological therapies according to gender (ITEM #4: “Do you think that pharmacological therapies should be differentiated by gender?”), 79% of respondents (206 women and 186 men) recognized this as noteworthy, without differences between men and women. Men who answer “No” are 4.8% more than women. No distinctions were detected among medical specialties’ (Appendix A).

Concerning Evidence-Based Medicine (EBM) demonstrated differences of symptoms in myocardial infarction according to gender (ITEM #5: “Do you think there are gender differences (demonstrated by EBM) in the presentation of myocardial infarction symptoms?”), 55.4% of respondents recognized these differences: 154/275 women (56%) thought that the presentation of myocardial infarction symptoms depends on gender differences (on the demonstrated EBM basis) against 121/275 (44%) men. The most positive responders were women in Group III (82.4%) and men in Group I (75%). None of the respondents in Groups I and II answered “No” (Appendix A).

For the gender differences (EBM proven) in smoking cessation strategies (ITEM #6: “Have you ever used different strategies in the treatment of cigarette smoking cessation, as stated by gender differences (demonstrated by EBM)?”), 21.6% of respondents (51 women and 56 men) reported they carried out differentiated strategies. The number of women who did not use differentiated strategies in the application of cigarette smoking cessation was 7.6% higher than the related percentage of men (178 vs. 153 respectively). No distinctions were detected among medical specialties (Appendix A).

Concerning the importance to develop educational programs on gender medicine (ITEM#7: “In your opinion, should the training of physicians include transversal specific topics on sex and gender differences?”), 88.5% of the respondents (231 women, 208 men) believed that training medical programs should include cross-culturally specific topics on gender differences, without particular differences among medical specialty Groups, except for all women (100%) of Group III e IV who responded ‘Yes’ (Appendix A).

As regards the willingness to follow educational courses or participate in events dedicated to gender medicine (ITEM #8: “Would you like to take courses or events to increase your knowledge on this topic?”), 360 respondents out of 496 (72.6%) would like to take courses to increase their knowledge on this topic with the percentage of women 13.4% higher than that of men. In contrast, 30 respondents replied ‘No’: 11 Women (36.7%) and 19 Men (63.3%) (Appendix A).

The last question (ITEM #9: “If no, why?”) was dedicated to the 30 who replied “No” to ITEM#8. On a total of 9 (A) answers six were men (66.7%) who would not like to take courses or events on this topic because they did not believe in gender medicine. Instead, only 3 out of 19 men believed that this training mode is not the best way to update. Half of the women belonging to Group VI e VII, replied ‘Other’ to explain the answer ‘No’ to ITEM#8. Half of the men belonging to Groups III, IV and VII do not believe in gender medicine, while the remaining men, 50% of Groups VI and VII stated to be aware of sex and gender differences in medicine (Appendix A).

## 4. Discussion

Medicine, from its origins, has had an androcentric attitude, relegating interests in women’s health to specific aspects related to reproduction. Since the 90s, traditional western medicine has undergone a profound evolution through an innovative approach aimed at studying the impact of gender and all the variables that characterize it (i.e., biological, environmental, social, economic, and cultural) on the physiology, pathophysiology, and clinical peculiarities of the diseases [10]. Briefly, from this perspective, the study of women’s health is no longer related to exclusively female diseases that affect principally the breast, uterus, and ovaries, but it considers the fact that the woman is not a copy of a man forward to the concept of “patient-centeredness” and of personalization of therapies”. In our exploratory study, 71.16% of the physicians answered to know the field of interests of gender medicine, indicating that more than one-quarter of the respondents has never been reached by informational messages in any form (e.g., printed, media, web) on the topic.

Several disparities in diagnosis, therapy, and outcomes have been motivated by the lack of investigations in female animals and women. The rareness of EBM results has created a bias that particularly affects women, who have historically been neglected in clinical research except for studies on the procreative apparatus [11,12]. Regarding the pharmacological responses of human beings, it depends on numerous factors and their reciprocal relationships, and it has been known that sex and gender differences imply drug consumption and adherence to therapy. Another factor recently investigated is the intestinal microbiota, conditioned by sex–gender differences, able to mitigate drug side effect, affects drug efficacy, and control antibiotic resistance [12,13,14].

Three items were proposed to the physicians about this specific topic: in our study, 87.9% of respondents recognized that knowing the differences between sex and gender gets better the healthcare ability, but strangely (incoherently) only 41.93% of respondents recognized that clinical trials were designed on male sex, and 79.03% of respondents recognized the importance of differentiating pharmacological therapies according to sex. These results suggest that our study group does not identify as a cornerstone of current clinical practice the fact that the trials were performed only on men and that the current administered drug therapies do not meet the principle of gender medicine (that responders invoked as positive and relevant). In Italy, mortality from cardiovascular diseases (CVDs) (cardiac and cerebral) is 48.4% in women and 38.7% in men [15].

The leading cause of death in women in all industrialized countries is myocardial infarction although heart failure has different characteristics in women and its impact effects are greater in old age women than men. Also, stroke affects women more than men and the ischemic form is more frequent and affects more women than men [16,17]. The prevalent risk factors for stroke in women are several: increased platelet reactivity, higher level of coagulation factors, and sex-associated unique cardiovascular risk factors, such as pregnancy-related (i.e., pre-eclampsia and gestational diabetes), gynecological disorders (i.e., polycystic ovary syndrome, early menopause) and autoimmune or systemic inflammatory diseases [18]. In Italy, stroke represents the third leading cause of death in men and the second in women [16]. There is a growing awareness that sex disparities in acute cardiovascular care (e.g., acute myocardial infarction, cardiogenic shock, cardiac arrest, mechanical circulatory support) are not new and are pervasive as reported by Vallabhajosyula et al. [19]. A very low result comes out of this study, since only 55.44% of respondents recognized these critical and huge differences for a correct workflow, for example, at the emergency room, confirming the disappointing picture that the Authors have drawn.

Tobacco use represents a significant factor of health risk [20]. Worldwide, 31% of men and 6% of women, with about 6 million related deaths. In most European Regions of the World Health Organization (WHO) countries, the prevalence of smokers varies between 21% and 30% [21]. Overall, about 41% of men and 22% of women smoke; in adolescents, the gender differences are minor: 20% of men and 15% of women 13–15 years old [22]. Tobacco uses in women, young and adult, therefore appears to be a behavior that needs to be carefully monitored and addressed. The issues in the female gender are related mainly to certain aspects of the habit: consumption, exposure to secondhand smoke from male smokers, and the use of household resources for the purchase of tobacco products rather than other goods or services [23]. In Italy, tobacco smoking is the third leading cause of loss of life years due to disability, illness, or premature death, after malnutrition and hypertension. Smokers 15 years of age or older are 22%, an intermediate value in the European scenario. Tobacco smoking habits are rather homogeneous across the country, with the prevalence lower for men in the North and women in the South. The health consequences of smoking in Italy account in 2010 for more than 71,000 deaths (53,000 men–18,000 women), accounting for 12.5% of total mortality, down from 15.1% in 1998 [24,25]. Women have been found to have a harder time abandoning smoking than men. Women process nicotine, the addictive ingredient in tobacco, faster than men. Differences in metabolism may help explain why nicotine replacement therapies, like patches and gum, work better in men than in women, and men appear to be more sensitive to nicotine’s effects with respect to addiction. On the contrary, women may be more susceptible than men to non-nicotine factors, such as sensory and social stimuli associated with smoking [26]. When counseling smokers, it has been found that in women the focus should be on perceived internal problems, contrary to more external obstacles in men and women experience stronger cravings than men in response to stress [27]. Women could have trouble in tobacco quitting because of the possible post-cessation weight gain, a concern that should be addressed in behavioral counseling [28]. Furthermore, female smokers seem to prefer, differently from male smokers, non-coercive interventions (e.g., a group intervention offering support and positivity) [29]. In this exploratory study, the result for this item is the worst, only 21.57% of respondents reported having experiences with different strategies, revealing the quite scarce knowledge on this bio-psychological aspect, and the urgency to disseminate sex-gender differences in smoking habit and cessation techniques [29]. Postgraduate educational programs are considered essential in the view to realize a new deal for the application of gender medicine. In terms of general aims, it seems fundamental (1) to coordinate awareness-raising, training and refresher courses for health workers to disseminate, throughout the regional territory, policies on gender health that take into account the biological, environmental, cultural, psychological and socio-economic variables determined by gender in consideration of their impact on the physiology, pathology, the clinical characteristics of several diseases in all organ systems, (from cardiology to oncology and neurology) including oral soft tissues and dental diseases [30,31,32]; (2) to disseminate the culture of patient-centeredness and individualized treatment in the practice multidisciplinary medicine, in the areas of prevention, diagnosis, treatment and rehabilitation; and (3) to consolidate the national networks in gender medicine to counter inequalities caused by the disallowance of the health impact of the gender-specific determinants. In this exploratory study, the percentage of physicians who believe that gender medicine should be taught to physicians is very high (88.5%), whereas for those willing to follow a course is a slightly lower percentage (77.6%).

To date, no other exploratory studies have been conducted about the knowledge and needs of the physicians in Italy on the actual topic of gender medicine, for this reason, it is not possible to indicate whether the low response rate achieved by the present study (496/8023; 6.18%) can be considered acceptable. Despite the certainty of anonymity, some participants may have erroneously reported their knowledge of gender medicine. Considering that the data collected were self-reported (some retrospective in nature) and since the questionnaire was not previously validated, response biases may limit the internal and external validity of the achieved results. It is desirable to validate this used explorative questionnaire and select a larger representative sample that includes a greater number of Sicilian physicians, extending to eastern Sicily and possibly to the Italian national territory for future insights.

## 5. Conclusions

Physicians seem to know the importance of gender medicine principles, embracing such a subject matter, although their knowledge on some very hot and renowned medical gender issues. The results of this first exploratory study confirm the need for specific training on good practices in gender medicine. Beyond the general awareness of doctors, therefore there is a mandatory need for dedicated resources and ad hoc training programs for both pre-graduate and post-graduate healthcare professionals. In this context, a new perspective will characterize not the birth of a new specialty but a better medical specificity.

The present survey should encourage facing the gender medicine gap in the current curricula of health professionals and, in this context, the transitional value of sex and gender principles in a clinical setting should be implemented.

## Figures and Tables

**Table 1 ijerph-20-00827-t001:** Group of medical specialties clustered by affinity. Group I: Immunological and oncological specialties; Group II: Neuro-psychiatric specialties; Group III: Internal and radiological specialties; Group IV: Gynecological and pediatric specialties; Group V: Sensory organs specialties; Group VI: Public and family specialties; Group VII: Miscellaneous.

Group	Medical Specialties
*Group* * I *	Rheumatology, Allergology and Clinical immunology, Oncology
*Group* * II *	Neurology, Child and adolescent psychiatry, Psychiatry
*Group* * III *	Endocrinology and diabetology, Gastroenterology, Pulmonology, Radiology, Urology, Nephrology, Cardiology, Internal medicine
*Group* * IV *	Pediatrics, Obstetrics and gynecology, Neonatology
*Group* * V *	Audiology and phoniatrics, Otorhinolaryngology, Ophthalmology, Dentistry, Stomatology
*Group* * VI *	Geriatrics, Family and general medicine (general practitioners), Epidemiology and public health, Medical guard, Health services organization, Community medicine, Without specialty
*Group* * VII *	Palliative care, Infectious diseases, Dermatology and venereology, Orthopedics and traumatology, Physical medicine and rehabilitation, Sports medicine, Food science and dietetics, Emergency medicine and surgery, Legal medicine, Occupational medicine, Intensive care medicine, Maxillofacial surgery, Plastic and reconstructive surgery, General surgery

**Table 2 ijerph-20-00827-t002:** Questions of the online questionnaire.

Item #	Questions and Answers
**ITEM #1**	** *Do you know what gender medicine deals with?* **
	□ No, □ Yes,
**ITEM #2**	** *In clinical practice, does the knowledge of the differences between sex and gender improve the ability to treat patients?* **
	☐ No, ☐ Maybe, ☐ Yes,
**ITEM #3**	** *In your opinion, does most of the medical knowledge come from studies conducted mainly on men?* **
	☐ No ☐ I do not know, ☐ Yes
**ITEM #4**	** *Do you think that pharmacological therapies should be differentiated by gender?* **
	☐ No, ☐ I do not know, ☐ Yes
**ITEM #5**	** *Do you think there are gender differences (demonstrated by EBM) in the presentation of myocardial infarction symptoms?* **
	☐ No, ☐ I do not know, ☐ Yes
**ITEM #6**	** *Have you ever used different strategies in the treatment of cigarette smoking cessation, as stated by gender differences (demonstrated by EBM)?* **
	☐ No, ☐ I do not know, ☐ Yes
**ITEM #7**	** *In your opinion, should the training of doctors include specific transversal topics on sex and gender differences?* **
	☐ No, ☐ I do not know, ☐ Yes
**ITEM #8**	** *Would you like to take courses or events to increase your knowledge on this topic?* **
	☐ No, ☐ Maybe, ☐ Yes,
**ITEM #9(*8.1*)**	** *If 8.1 is No, why?* **
	☐ I do not believe in gender medicine (A)☐ I do not believe this is the best way for my update (B)☐ I am aware of the differences between sex and gender in medicine (C)☐ Other (D)

**Table 3 ijerph-20-00827-t003:** Details of participants by medical specialty group, age, and gender.

*Group*	*n* %	*Women*		*Total*	*Men*		*Total*	*Total*
		Age ≤ 50	Age >50		≤50	>50		
** *Group I* **	** *n* **	**4**	**3**	**7**	**2**	**2**	**4**	**11**
	**%**	57.1	42.9	63.6	50	50	36.4	2.2
** *Group II* **	** *n* **	**9**	**16**	**25**	**0**	**6**	**6**	**31**
	**%**	36	64	80.6	0	100	19,4	6,3
** *Group III* **	** *n* **	**10**	**7**	**17**	**6**	**24**	**30**	**47**
	**%**	58.8	41.2	36.2	20	80	63.8	9.5
** *Group IV* **	** *n* **	**6**	**9**	**15**	**2**	**21**	**23**	**38**
	**%**	40	60	39.5	8.7	91.3	60.5	7.7
** *Group V* **	** *n* **	**5**	**4**	**9**	**3**	**14**	**17**	**26**
	**%**	55.6	44.4	34.6	17.6	82.4	65.4	5.2
** *Group VI* **	** *n* **	**56**	**78**	**134**	**29**	**95**	**124**	**258**
	**%**	41.8	58.2	51.9	23.4	76.6	48.1	52
** *Group VII* **	** *n* **	**20**	**34**	**54**	**8**	**23**	**31**	**85**
	**%**	37	63	63.5	25.8	74.2	36.5	17.1
** *TOTAL* **	** *n* **	**110**	**151**	**261**	**50**	**185**	**235**	**496**
	**%**	**42.1**	**57.9**	**52.6**	**21.3**	**78.7**	**47.4**	**100**

**Table 4 ijerph-20-00827-t004:** Answers of respondents to the 9 items, differentiated by gender (* only for responders ‘No’ to item 8).

*ITEM#*	*Answer*	*TOTAL* *N. (Answers/Total Responders) (%)*	*Women* *N. (Answers/Total Women Responders) (%)*	*Men* *N. (Answers/Total Men Responders)* *(%)*
** *ITEM#1* **	**No**	**143** (28.8)	**81** (56.6)	**62** (43.4)
	**Yes**	**353** (71.2)	**180** (51)	**173** (49)
** *ITEM#2* **	**No**	**15** (3)	**8** (53.3)	**7** (46.7)
	**Maybe**	**45** (9)	**29** (64.4)	**16** (35.6)
	**Yes**	**436** (88)	**224** (51.4)	**212** (48.6)
** *ITEM#3* **	**No**	**144** (29)	**75** (52.1)	**69** (47.9)
	**I don’t know**	**144** (29)	**76** (52.8)	**68** (47.2)
	**Yes**	**208** (42)	**110** (52.9)	**98** (47.1)
** *ITEM#4* **	**No**	**77** (15.6)	**37** (47.1)	**40** (51.9)
	**I don’t know**	**27** (5.4)	**18** (67.7)	**9** (33.3)
	**Yes**	**392** (79)	**206** (52.5)	**186** (47.5)
** *ITEM#5* **	**No**	**91** (18.4)	**36** (39.6)	**55** (60.4)
	**I don’t know**	**130** (26.2)	**71** (54.6)	**59** (45.4)
	**Yes**	**275** (55.4)	**154** (56)	**121** (44)
** *ITEM#6* **	**No**	**331** (66.7)	**178** (53.8)	**153** (46.2)
	**I don’t know**	**58** (11.7)	**32** (55.2)	**26** (44.8)
	**Yes**	**107** (21.6)	**51** (47.7)	**56** (52.3)
** *ITEM#7* **	**No**	**29** (5.9)	**16** (55.2)	**13** (44.8)
	**I don’t know**	**28** (5.6)	**14** (50)	**14** (50)
	**Yes**	**439** (88.5)	231 (52.6)	**208** (47.4)
** *ITEM#8* **	**No**	**30** (6)	**11** (36.7)	**19** (63.3)
	**Maybe**	**106** (21.4)	**46** (43.4)	**60** (56.6)
	**Yes**	**360** (72.6)	**204** (56.7)	**156** (43.3)
** *ITEM#9 ** **	**(A)**	**9/30 *** (30)	**3** (33.3)	**6** (66.7)
	**(B)**	**5/30 *** (16.7)	**2** (40)	**3** (60)
	**(C)**	**6/30 *** (20)	**1** (16.7)	**5** (83.3)
	**(D)**	**10/30 *** (33.3)	**5** (50)	**5** (50)

## Data Availability

Data available on request.

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
