# Peer review of "Gender and Sex in Medical Practice: An Exploratory Study on Knowledge, Behavior, and Attitude among Sicilian Physicians"

_ijerph, 2023, doi:10.3390/ijerph20010827_

Round 1
Reviewer 1 Report
This work is also interesting because the investigation takes place in an area of ​​southern Europe often overlooked in relation to the northern hemisphere. English language should be revised by a native English speaker because some are quite hard to understand. Through the paper you speak of gender medicine, gender and sex specific medicine and gender and sex medicine please use only a term because they are not synonyms. I suggest to use the term men and women instead of males and females because these last 2 terms refer only to biological differences Introduction Line 32, I suggest to remove specific after sex and gender because at an international level not everyone agrees to name it in this way while the. Line 32, actually it is well known that also autosomal genes are implicate in sex differences substitute sex chromosomes with (genes and hormones).From line 35 to 37 and From line 41 to 44 add more citations
Line 47-67 this part it is too long and unclear, it must be rewritten and shortened. Methods Please briefly described how you clustered medical discipline Results differentiated for sex should be for gender Discussion It is too long andthe discussion is too long and it is not focused on the interesting data that emerge from result section Indeed, Data found describe a very varied landscape on medical interest and knowledge of Sicilian physicians building the base for implementing the awareness on gender and sex medicine.
Author Response
Response to Reviewer 1 Comments
Review report:
This work is also interesting because the investigation takes place in an area of southern Europe often overlooked in relation to the northern hemisphere. English language should be revised by a native English speaker because some are quite hard to understand. Through the paper you speak of gender medicine, gender and sex specific medicine and gender and sex medicine please use only a term because they are not synonyms. I suggest to use the term men and women instead of males and females because these last 2 terms refer only to biological differences Introduction Line 32, I suggest to remove specific after sex and gender because at an international level not everyone agrees to name it in this way while the. Line 32, actually it is well known that also autosomal genes are implicate in sex differences substitute sex chromosomes with (genes and hormones).
From line 35 to 37 and From line 41 to 44 add more citations
Line 47-67 this part it is too long and unclear, it must be rewritten and shortened. Methods Please briefly described how you clustered medical discipline Results differentiated for sex should be for gender Discussion It is too long and
the discussion is too long and it is not focused on the interesting data that emerge from result section Indeed, Data found describe a very varied landscape on medical interest and knowledge of Sicilian physicians building the base for implementing the awareness on gender and sex medicine.
Response.
Thank you for kind comments.
As required, the correct term “gender medicine” has been applied throughout the entire text and the terms "males" and "females" have been replaced by "men" and "women".
Thanking you for the advice and related information provided, on line 32 we have removed the term "specific" and substituted “sex chromosomes” with “genes and hormones”. Citations have been added as needed in lines 35-37 and 41-44 and the section between rows 47-67 has been rewritten, shortened, and made easier to understand as necessary and clear.
A brief description of the clustering of medical specialties has been added in the text and for each cluster, further criteria are reported in the caption of the Table 1.
Based on the questionnaire used, the discussions were suitably guided by the various landscape on medical knowledge of Sicilian physicians we found in descriptive obtained results. This first exploratory study carried out in a provincial area of southern Italy will allow, as you suggest, to build the base for implementing the awareness on gender and sex medicine at higher, regional, and national levels.
As suggested by Editors, the manuscript will undergo to extensive English revisions after second peer-review using IJERPH editing services (https:///www.mdpi.com/authors/english).

Reviewer 2 Report
Descriptive statistics alone may not be worth publishing as a paper with this content.
The paper should be written in a way that follows the usual procedure of first setting up a hypothesis, conducting a statistical analysis on it, and then showing that the hypothesis is significantly supported or rejected.
The data obtained are considered to be sufficient for such an analysis, so it is advisable to consult a statistician to improve the content.
Author Response
Response to Reviewer 2 Comments
Review report:
Descriptive statistics alone may not be worth publishing as a paper with this content.
The paper should be written in a way that follows the usual procedure of first setting up a hypothesis, conducting a statistical analysis on it, and then showing that the hypothesis is significantly supported or rejected.
The data obtained are considered to be sufficient for such an analysis, so it is advisable to consult a statistician to improve the content.
Response
Thank you for your kind suggestions.
The study was designed as an exploratory survey aimed to investigate the knowledge of sex and gender in clinical medicine among Sicilian physicians. It is the first investigative study on the awareness of this psychological and actual multidimensional subject in the medical clinical setting, made in this southern Italian region. The questionnaire and the successive statistical analysis were specifically designed and modeled by an epidemiologist statistician (Prof. Massimo Attanasio, ORCID: https://orcid.org/0000-0003-2684-5530) after a meticulous analysis of the literature.
The data analysis consequently was conducted considering the data set obtained from the questionnaires for preliminary and novel descriptive results.
As this study is an exploratory study proposed by the Medical Council of Palermo, the actual aim of the study is evaluating the effective knowledge on topic and enhance the future economic funds and human resources, sampling the target population of Sicilian Physicians, starting intentionally from a descriptive analysis.
We consider the study approach you proposed a best applicable for the data type; certainly, to be applied to a wider sample of Sicilian doctors (with extension to eastern Sicily).
However, given the exploratory nature of this work, we believe that the description of the data proposed can provide a substrate for future planning.
As suggested by Editors, the manuscript will undergo to extensive English revisions after second peer-review using IJERPH editing services (https:///www.mdpi.com/authors/english).

Reviewer 3 Report
I congratulate the authors for addressing an issue of great current interest and one that needs to be addressed. Gender bias in health care places women, because they are women, in a situation of vulnerability and inequality in health.
METHODS SECTION
STUDY DESIGN
The classificatory criterion used in the grouping of specialities for data collection is not reflected.
QUESTIONNAIRE
The questionnaire used for data collection does not appear to have been previously validated, so there may be biases that condition the results. An explanation in this respect is required from the authors of the article.
The approval of the ethics committee should not be reflected in this section, it requires its own section "Ethical considerations".
RESULTS
The participation of professionals was very low, something already acknowledged by the authors of the study. I doubt that this sample size will yield statistically significant results for the analysis of the variables considered.
DISCUSSION
It would be desirable to expand this section by incorporating into the discussion other studies that have been carried out. The scientific literature on the subject covered in this study is extensive and up to date.
Author Response
Response to Reviewer 3 Comments
Review report:
I congratulate the authors for addressing an issue of great current interest and one that needs to be addressed. Gender bias in health care places women, because they are women, in a situation of vulnerability and inequality in health.
METHODS SECTION
STUDY DESIGN
The classificatory criterion used in the grouping of specialties for data collection is not reflected.
Response.
Thank you for your kind comment. With respect to the clustering of medical specialties the classificatory criteria have been reported in the text as required; for each cluster, further grouping criteria are reported in the caption of the Table 1.
QUESTIONNAIRE
The questionnaire used for data collection does not appear to have been previously validated, so there may be biases that condition the results. An explanation in this respect is required from the authors of the article.
The approval of the ethics committee should not be reflected in this section, it requires its own section "Ethical considerations".
Response.
Thank you for your comments and useful suggestions.This study is the first investigative evaluation on the awareness of a psychological and actual multidimensional subject in the medical clinical setting, made in this southern Italian region. The questionnaire and the successive statistical analysis were specifically designed and modeled by an epidemiologist statistician (Massimo Attanasio, ORCHID: https://orcid.org/0000-0003-2684-5530) after a meticulous analysis of the literature on the subject and an application of the contents studied in the context of reference. The data analysis consequently was conducted considering the data set obtained from the questionnaires for preliminary and novel descriptive and explorative results. As you affirm, we consider the questionnaire validation a correct applicable approach for the study. We are aware that considering that the questionnaire was not previously validated, response biases may limit the validity of the achieved results and this possibility is specifically expressed in the final discussion section.
For this reason, it is essential and desirable to select a larger representative sample that includes a greater number of Sicilian physicians to allow the validation of this questionnaire that now is purely exploratory and extend its administration to the remaining part of the Italian national territory.
Finally, as you evidenced, the approval of the ethics committee has been transferred in a new paragraph of “Ethical consideration” with related details.
RESULTS
The participation of professionals was very low, something already acknowledged by the authors of the study. I doubt that this sample size will yield statistically significant results for the analysis of the variables considered.
Response.
Thank you for this observation. As you highlighted, the participation of physicians to the response to the questionnaire was low in the specific tested sample. Given the innovativeness of the research under examination and given the investigative, descriptive, and especially exploratory nature, it is already fundamental to highlight how the sample response is really low. Just this data is an important result of our study and will be the focus for further research and future insights. Thank to this study we obtained considerable results that, as you pointed out, is addressed to an issue of great current interest, particularly since the study was conducted in a southern Italian region where the social inequality of women is strongly present.
DISCUSSION
It would be desirable to expand this section by incorporating into the discussion other studies that have been carried out. The scientific literature on the subject covered in this study is extensive and up to date.
Response.
Thank you for your useful suggestions, to expand the discussion other carried out studies have been incorporated in the text and cited.
As suggested by Editors, the manuscript will undergo to extensive English revisions after second peer-review using IJERPH editing services (https:///www.mdpi.com/authors/english).

Round 2
Reviewer 2 Report
I think that the revised manuscript has no major problems and is acceptable for publication.